# Material Properties of Zr–Cu–Ni–Al Thin Films as Diffusion Barrier Layer

**Po-Hsien Sung and Tei-Chen Chen \***

Department of Mechanical Engineering, National Cheng Kung University, Tainan 70101, Taiwan;
Pohsien_Sung@aseglobal.com
\* Correspondence: ctcx831@mail.ncku.edu.tw

**Abstract:** Due to the rapid increase in current density encountered in new chips, the phenomena of thermomigration and electromigration in the solder bump become a serious reliability issue. Currently, Ni or TiN, as a barrier layer, is widely academically studied and industrially accepted to inhibit rapid copper diffusion in interconnect structures. Unfortunately, these barrier layers are polycrystalline and provide inadequate protection because grain boundaries may presumably serve as fast diffusion paths for copper and could react to form Cu–Sn intermetallic compounds (IMCs). Amorphous metallic films, however, have the potential to be the most effective barrier layer for Cu metallization due to the absence of grain boundaries and immiscibility with copper. In this article, the diffusion properties, the strength of the interface between polycrystalline and amorphous ZrCuNiAl thin film, and the effects of quenching rate on the internal microstructures of amorphous metal films were individually investigated by molecular dynamics (MD) simulation. Moreover, experimental data of the diffusion process for three different cases, i.e., without barrier layer, with an Ni barrier layer, and with a $Zr_{53}Cu_{30}Ni_9Al_8$ thin film metallic glass (TFMG) barrier layer, were individually depicted. The simulation results show that, for ZrCuNiAl alloy, more than 99% of the amorphous phase at a quenching rate between 0.25 K/ps and 25 K/ps can be obtained, indicating that this alloy has superior glass-forming ability. The simulation of diffusion behavior indicated that a higher amorphous ratio resulted in better barrier performance. Moreover, a very small and uniformly distributed strain appears in the ZrCuNiAl layer in the simulation of the interfacial tension test; however, almost all the voids are initiated and propagated in the Cu layer. These phenomena indicate that the strength of the ZrCuNiAl/Cu interface and ZrCuNiAl layer is greater than polycrystalline Cu. Experimental results show that the $Zr_{53}Cu_{30}Ni_9Al_8$ TFMG layer exhibits a superior barrier effect. Almost no IMCs appear in this TFMG barrier layer even after aging at 125 °C for 500 h.

**Keywords:** thin film metallic glasses (TFMGs); ZrCuNiAl; diffusion barrier layer; molecular dynamics (MD); microbump; intermetallic compounds (IMCs)

---

## 1. Introduction

In the current semiconductor device industry, 2.5-dimensional (2.5D) integrated circuit (IC) flip chip assembly packages with microbump are widely used for high-end niche applications [1]. On the other hand, copper is gradually replacing aluminum as a lead and trace material due to 40% lower resistance than aluminum. However, the major drawback of using copper as metallization is that copper has a diffusion coefficient much higher than aluminum [2]. In addition, copper and tin can easily react to form Cu–Sn intermetallic compounds (IMCs) at temperatures even as low as 200 °C. Although the formation of an IMC layer signifies good bonding between the solder and Cu pad, it becomes the most brittle part in the solder joint and, thus, easily results in premature failure of electronic devices caused by excessive growth of IMCs. Hence, the issue of selecting a proper material to act as a diffusion

barrier between copper and tin to prevent interaction is becoming more important in the trend of minimization of electronics devices. In recent years, materials of TiN [3–6], Ni [7–9], and Ta [10] as barrier layers were widely investigated, and they are industrially accepted to inhibit rapid copper diffusion in interconnect structures. Unfortunately, TiN and Ni barrier layers are polycrystalline and provide inadequate protection because grain boundaries may presumably serve as fast diffusion paths for copper and could react to form Cu–Sn IMCs. In particular, when the bump size dimensions become small toward 30 μm and below, IMCs tend to occupy a large volume ratio in the solder bumps as shown in Figure 1. In addition, the current density in the solder bump needs to be further increased. Under such high current density, the thermomigration and the electromigration in the solder bump become a serious reliability issue [11–14]. Prevention of the growth of IMCs continues to be a critical limitation for improving the performance of integrated circuits.

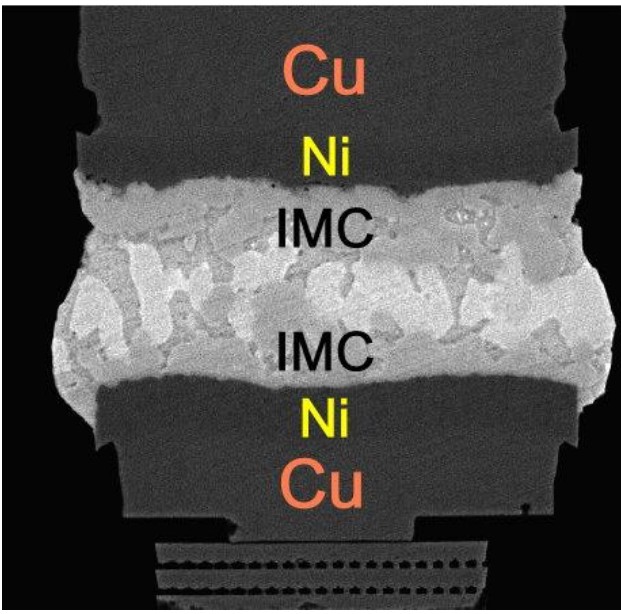

**Figure 1.** Cross-section of a microbump in a 2.5-dimensional (2.5D) integrated circuit (IC) package.

Thin film metallic glasses (TFMGs) are regarded as excellent diffusion barriers in integrated circuit applications [15]. In particular, refractory metal nitrides, deposited by reactive sputtering, were shown to be promising diffusion barriers. These kinds of barriers block the interaction between the interconnect materials and the silicon, and they are potentially useful as a diffusion barrier in other systems, in addition to those between the interconnect materials and the silicon [16,17]. Recently, TFMGs were gradually recognized and accepted as the most effective barriers layer for Cu metallization due to the absence of grain boundaries and immiscibility with copper [18–20]. Since the diffusion coefficient for Sn in Cu is 10 orders of magnitude smaller than that for Cu in Sn [21], IMCs are scarcely found and grown in Sn. As long as the diffusion of Cu from the Cu pad into the barrier layer can be effectively reduced, it represents that this barrier layer can also slow down the IMC growth. Zr–Cu–Ni–Al bulk amorphous alloy is a unique material due to its superior glass-forming ability and high strength [22]. As a potential candidate of TFMGs, its diffusion behavior and interface strength with the Cu layer are crucial and need further investigation. In this study, the material properties, especially diffusion behavior and strength, of ZrCuNiAl amorphous metal films were systematically examined using both numerical and experimental methods. Some important findings are illustrated in detail. The overall schematic diagram of this work is depicted in Figure 2.

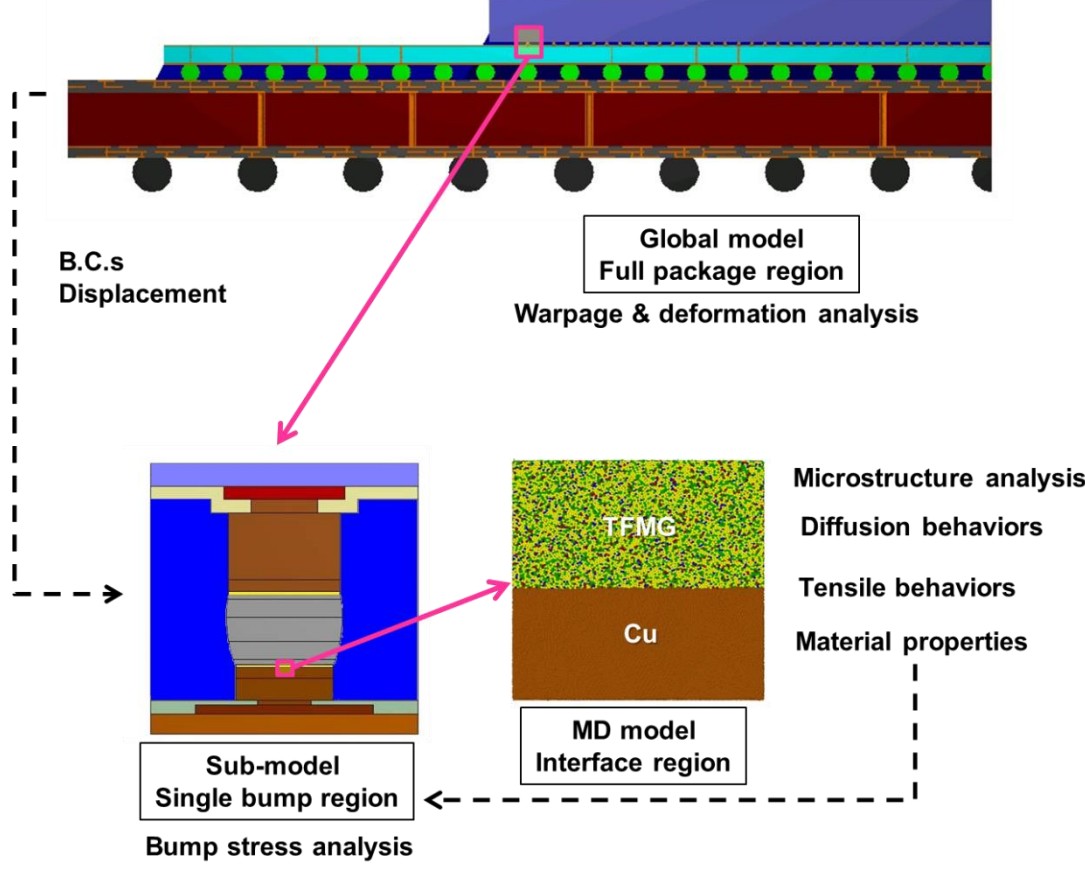

**Figure 2.** Overall schematic diagram of this study.

## 2. Methodology

### 2.1. Molecular Dynamics Simulation

#### 2.1.1. Amorphous Geometry Structure

The initial structure of the molecular dynamics (MD) simulation model was constructed using the Large-scale Atomic/Molecular Massively Parallel Simulator (LAMMPS) ( a free and open-source software from Sandia National Laboratories, distributed under the terms of the GNU General Public License at https://lammps.sandia.gov/) [23] and the Open Visualization Tool (OVITO) package (a free and open-source scientific visualization and analysis package for atomistic and particle-based simulation data at https://www.ovito.org/) [24]. The embedded-atom method (EAM) potential was adopted to model the atomistic interactions among Cu, Zr, Ni, and Al atoms [25,26]. This potential was successfully applied to simulate the structure, surface, and transformation of amorphous metallics. The amorphous structure of the Zr–Cu–Ni–Al layer at a temperature of 300 K was obtained by performing simulations with specific parameter settings and heat treatments. Firstly, a crystal Zr–Cu–Ni–Al alloy was created with a face-centered cubic copper lattice of 185,000 atoms and enough copper atoms were randomly replaced with Zr, Ni, and Al in disordered substitution to give the system the desired composition. The system was initially relaxed under periodic boundary conditions at 300 K for 200 ps within a canonical ensemble (NVP) and then heated from 300 K to 2200 K at a constant heating rate of $5 \times 10^{11}$ K/s to $5 \times 10^{13}$ K/s. To make the state of the system as natural as possible, the liquid system was relaxed for 20 ps at 2200 K. Finally, the system was quenched from 2200 K to 300 K at a quenching rate of 0.5 K/ps, followed by relaxation for 20 ps at 300 K. This quenching rate is fast enough to achieve an amorphous phase. Figure 3 shows the atomic configurations of the Zr–Cu–Ni–Al model prior to and after quenching.

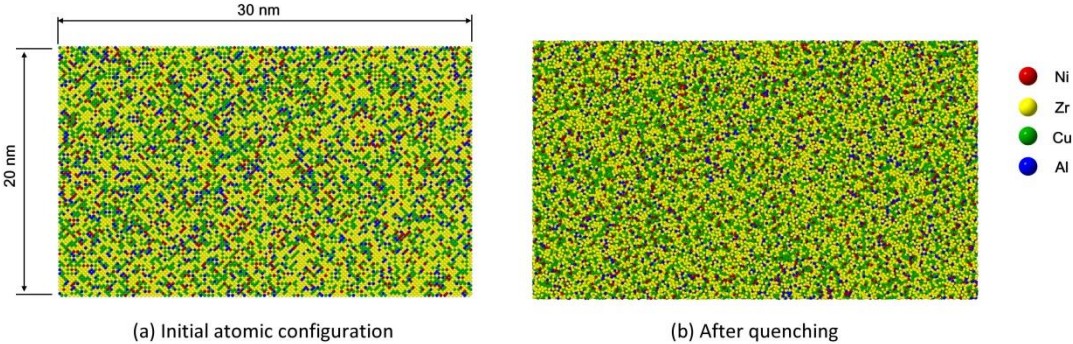

**Figure 3.** Atomic snapshots of Zr–Cu–Ni–Al model: (**a**) initial atomic configuration and (**b**) after quenching (quenched from 2200 K to 300 K at a quenching rate of 0.5 K/ps, followed by relaxation for 20 ps at 300 K).

### 2.1.2. Procedure for Interface Model Preparation

To obtain the Cu (polycrystalline)/Zr–Cu–Ni–Al (after quenching) interface model of size 30 nm width (*x*-axis) × 30 nm height (*z*-axis) × 5 nm thickness (*y*-axis), we started by constructing a Cu/Zr–Cu–Ni–Al crystalline model. The bottom layer was filled with Cu atoms (15 nm height) and the simulation box above it was filled with Zr–Cu–Ni–Al atoms. Initially, both Cu and Zr–Cu–Ni–Al layers were kept in contact along the interface in the *x*–*y* plane with a separation distance of 3.5 Å. The interface separation distance was chosen based on the equilibrium bond lengths. It was previously shown that, as long as the interface separation distance is not too small or not too large, it does not affect the results [27].

### 2.1.3. Procedure for Diffusion and Deformation

Models for diffusion and deformation are shown in Figure 4. Two additional rigid regions of dimensions 30 nm width (*x*-axis) × 5 nm height (*y*-axis) × 12 nm thickness (*z*-axis) were imposed for the latter at the top and bottom of the model. A uniform velocity in opposite direction, corresponding to a strain rate of $10 \times 10 \text{ s}^{-1}$, was individually applied to the top and bottom rigid regions along the *z*-axis direction. The tensile test was simulated at a temperature of 300 K under the NVT ensemble. In the numerical simulations, the derivatives in equations of motion were firstly predicted at time $t + \Delta t$ by applying Taylor expansions at $t$. Gear's Predictor–Corrector integration algorithms [28] were then used for the second-order differential equations to correct all predicated positions. The time integration of motion was performed with a time step of 2 fs.

The configuration of the system was firstly established; it was then relaxed by energy minimization to ensure no existence of steric clashes or inappropriate geometries. The convergence of the pressure, temperature, potential energy, and volume as a function of different times was conducted to ensure that the system reached a proper equilibrium, as shown in Figure 5. In these figures, the pressure and density have no significant change with time, and the energy potential and temperature gradually tend to stabilize when the time is greater than 100 ps. In other words, the simulation model can be presumed to be convergent, and it reaches a steady state. To verify the correctness of the intermolecular potential functions, Table 1 shows the elastic constants of calculated data by LAMMPS and experimental data [25,29–34]. It can be seen that, except for the elastic modulus $C_{44}$ of the Zr atom, the errors of other elements were within 5% between calculated data and experimental data. The error in $C_{44}$ might be attributed to the improper shearing model to predict $C_{44}$ or the incorrect parameterization for the hexagonal close packed (HCP) structure of the Zr atom. However, as shown in the literature [35,36], it should be emphasized that classical models are not always reliable and should be subjected to careful fitting to first-principle data. Furthermore, other more detailed verifications, such as atomic equilibrium distance, coefficient of thermal expansion, $T_g$ and $T_m$, and so on were individually performed and discussed.

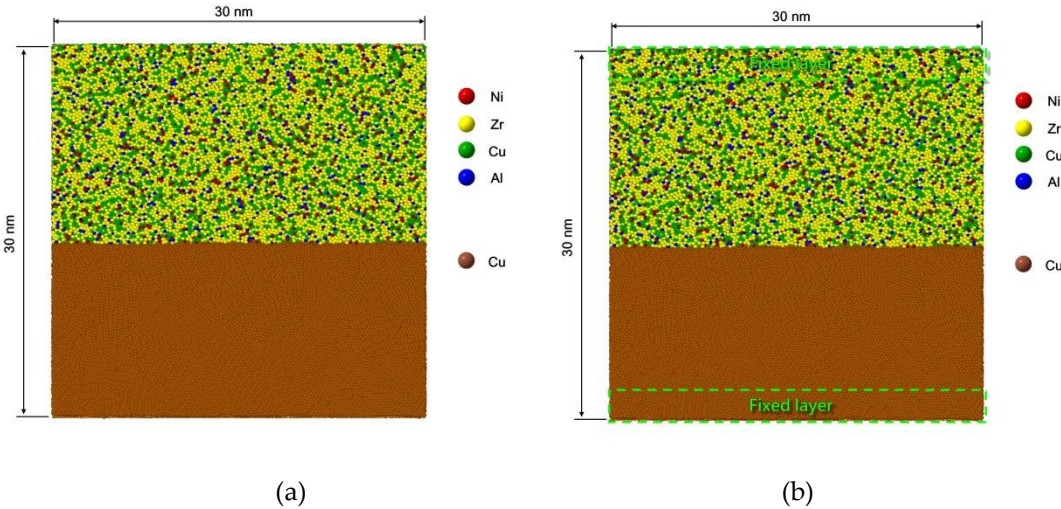

**Figure 4.** Atomic snapshots of Zr–Cu–Ni–Al model for (**a**) diffusion process and (**b**) tensile deformation process.

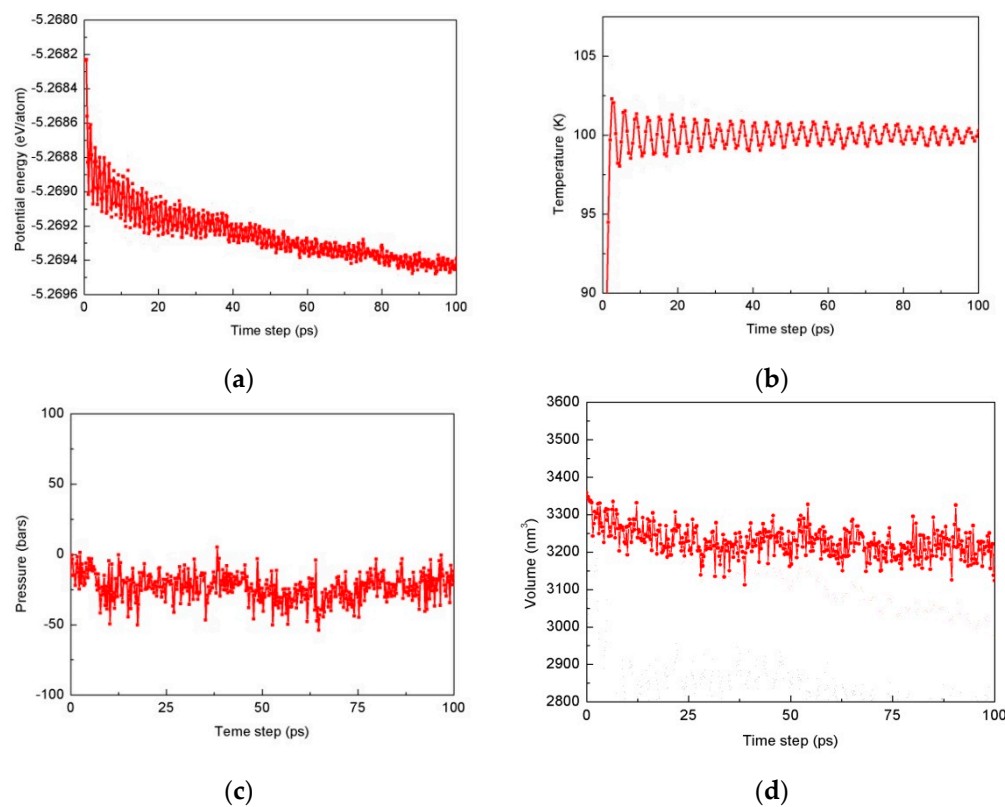

**Figure 5.** Graphs of the (**a**) potential energy minimization, (**b**) temperature, (**c**) pressure, and (**d**) volume equilibration of the Zr–Cu–Ni–Al alloy simulation system prior to simulation.

To analyze the thermodynamic properties of the Zr–Cu–Ni–Al alloy for different compositions and quenching rates, the relationship between volume and temperature was compared. Figure 6 shows the volume of 185,000 atoms versus temperature curves at different quenching rates of 0.25, 2.5, and 25 K/ps, which reveals the quenching rate dependence of glass transition temperature. In order to acquire a more refined estimate of glass transition temperature, the data were captured with smaller increments in temperature, i.e., 1 K, from 200 to 2000 K. During the quenching process, the

volume vs. temperature curve with a quenching rate of 25 K/ps showed a slope change at about 725 ± 30 K, which indicates that glass transition occurs at this temperature. Similar results were obtained in the experiments by Chang et al. [37] and Jang et al. [38]. As the quenching rate is within the range from 2.5 K/ps to 25 K/ps, there exists no significant difference in turning points due to no obvious crystalline structure at these quenching rates. The glass transition temperature of $T_g$ slightly decreases with the decrease in quenching rate. The result shows that a decrease in quenching rate causes an essential decrease in the volume at $T < T_g$, and the system is better equilibrated at a lower quenching rate. Figure 6 also shows that a higher quenching rate leads to a larger free volume in the final glassy state. When the quenching rate increased 10-fold from 0.25 to 2.5 K/ps, the volume of Zr–Cu–Ni–Al after quenching increased by approximately 1%. The predicted melting temperature of 1170 ± 30 K by MD calculations is in reasonable agreement with the experimental melting temperature of 1135 K [38]. In addition, the coefficient of thermal expansion (CTE) can be calculated through a volume–temperature diagram. The CTE of $Zr_{53}Cu_{30}Ni_9Al_8$ amorphous alloy in this work was about (12 ± 2) $\times 10^{-6}$/ K, similar to the experimental data reported in the literature [39,40]. In order to further understand the microstructures in amorphous structures, pair analysis techniques, especially the bond-type index method of Honeycutt–Anderson (HA) [41], were applied. The mean square displacement (MSD) profiles at temperatures ranging from 300 to 900 K for Cu–Ag and Zr–Cu–Ni–Al amorphous metal were used to investigate their dynamical properties.

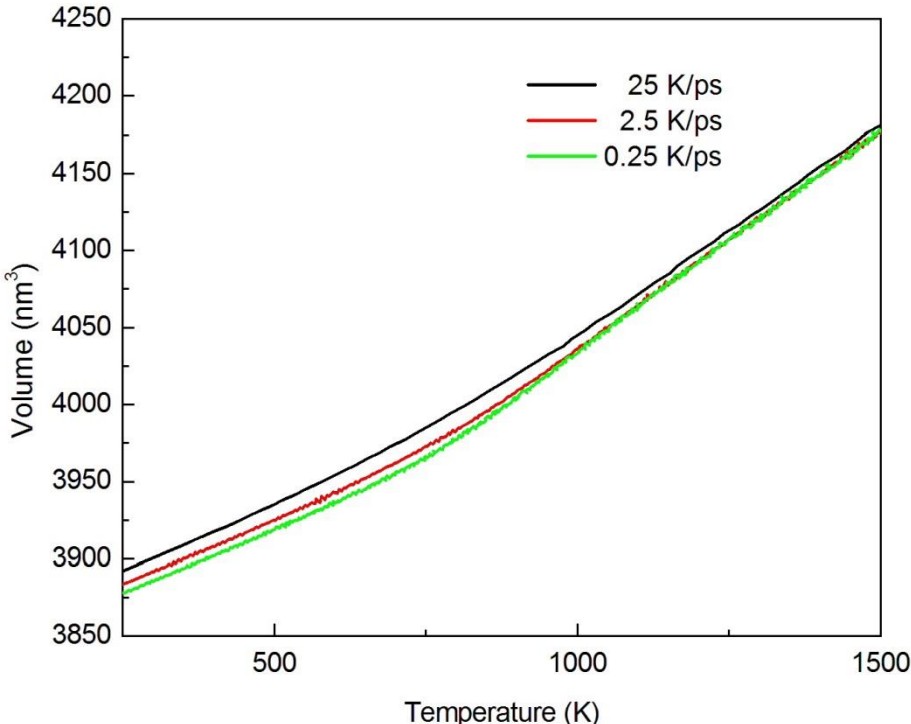

**Figure 6.** Volume variations of 185,000 atoms as a function of temperature under different quenching rates of $Zr_{53}Cu_{30}Ni_9Al_8$ alloy.

**Table 1.** Elastic constants of Cu, Zr, Ni, and Al determined experimentally and via molecular dynamics (MD).

| Element | Methods | $C_{11}$ (GPa) | $C_{12}$ (GPa) | $C_{44}$ (GPa) |
|---------|---------|--------------|--------------|--------------|
| Cu | Experimental data [25] | 168.4 | 121.4 | 75.4 |
|    | Calculated data by MD | 168.5 | 120.8 | 75.2 |
|    | Error | 0.06% | 0.49% | 0.27% |
| Zr | Experimental data [29] | 144 | 74 | 33.4 |
|    | Calculated data by MD | 141 | 74.2 | 44 |
|    | Error | 2.1% | 2.7% | 31.7% |
| Ni | Experimental data [25] | 253 | 152 | 124 |
|    | Calculated data by MD | 240.4 | 144.1 | 122.7 |
|    | Error | 4.98% | 5.20% | 1.05% |
| Al | Experimental data [25] | 107.3 | 60.1 | 28.3 |
|    | Calculated data by MD | 102.6 | 58.8 | 27.7 |
|    | Error | 4.38% | 2.16% | 2.12% |

*2.2. Sample Preparations and Experimental Verification*

To compare with the simulation results, the diffusion behaviors for three different samples, i.e., without a barrier layer, with an Ni barrier layer, and with a $Zr_{53}Cu_{30}Ni_9Al_8$ TFMG barrier layer, were individually investigated experimentally. The thickness of the Ni and $Zr_{53}Cu_{30}Ni_9Al_8$ TFMG barrier layers was 2.5 μm and 100 nm, respectively. A normal package test vehicle of 1.4 ✕ 1.4 mm was used as the substrate. The substrates were cleaned by acetone and ethanol. Prior to depositions of barrier layers, the organic solderability preservative (OSP) on the Cu pad can be properly removed by formic acid. ZrCuNiAl barrier layers were coated onto a Cu pad surface substrate by means of a vacuum DC MDX 1000 sputtering system (MeiVac Inc., San Jose, CA, USA). Sputtering involved firing ions toward a ZrCuNiAl bulk metallic glass (BMG) target to displace atoms, which were then deposited onto a Cu pad substrate to form a ZrCuNiAl thin film. A bare Cu pad substrate was used as a comparison. The schematic deposited structures for three kinds of samples of a full solder ball system, called a bump, were as shown in Figure 7. This full solder ball system is generally composed of a solder ball, Cu pad, barrier layer, and package substrate colored by green. Figure 7a represents the sample with no barrier layer, while Figure 7b,c show the sample with Ni and $Zr_{53}Cu_{30}Ni_9Al_8$ TFMG barrier layers, respectively. The operating conditions of the DC sputtering system were set at a base pressure of $6 \times 10^{-6}$ torr, a working pressure of 4 millitorr, argon (Ar) flow of 5.4 standard $cm^3$, sputtering time of 30 min, and varying sputtering powers of 15 W to 40 W.

In order to confirm the structure of the $Zr_{53}Cu_{30}Ni_9Al_8$ TFMG deposited by sputtering, this TFMG was investigated by scanning electron microscopy (SEM) (JEOL JSM-7000F) and energy-dispersive X-ray spectroscopy (EDS). The accelerating voltage for SEM/EDS was set to 5 kV, with a 30-s lifetime. The results of X-ray diffraction revealed that the $Zr_{53}Cu_{30}Ni_9Al_8$ alloy film exhibits a typical amorphous diffraction pattern. The ratio between Zr, Cu, Ni, and Al as an atomic percentage in the amorphous thin film was identified by SEM/EDS. The elemental composition data on the amorphous thin film are shown in Figure 8 and Table 2. The ratio between Zr, Cu, Ni, and Al as an atomic percentage in the amorphous thin film identified by SEM/EDS for $Zr_{51}Cu_{32}Ni_{10}Al_7$ was very close to the designed $Zr_{53}Cu_{30}Ni_9Al_8$ goal. When the barrier layers were deposited, the SAC305 solder balls of 300 μm diameter could be placed manually on the Cu pad and were held in place with a flux. Immediately, Cu pads of the substrate and the solder ball were joined together in an oven at 260 °C for 30 min.

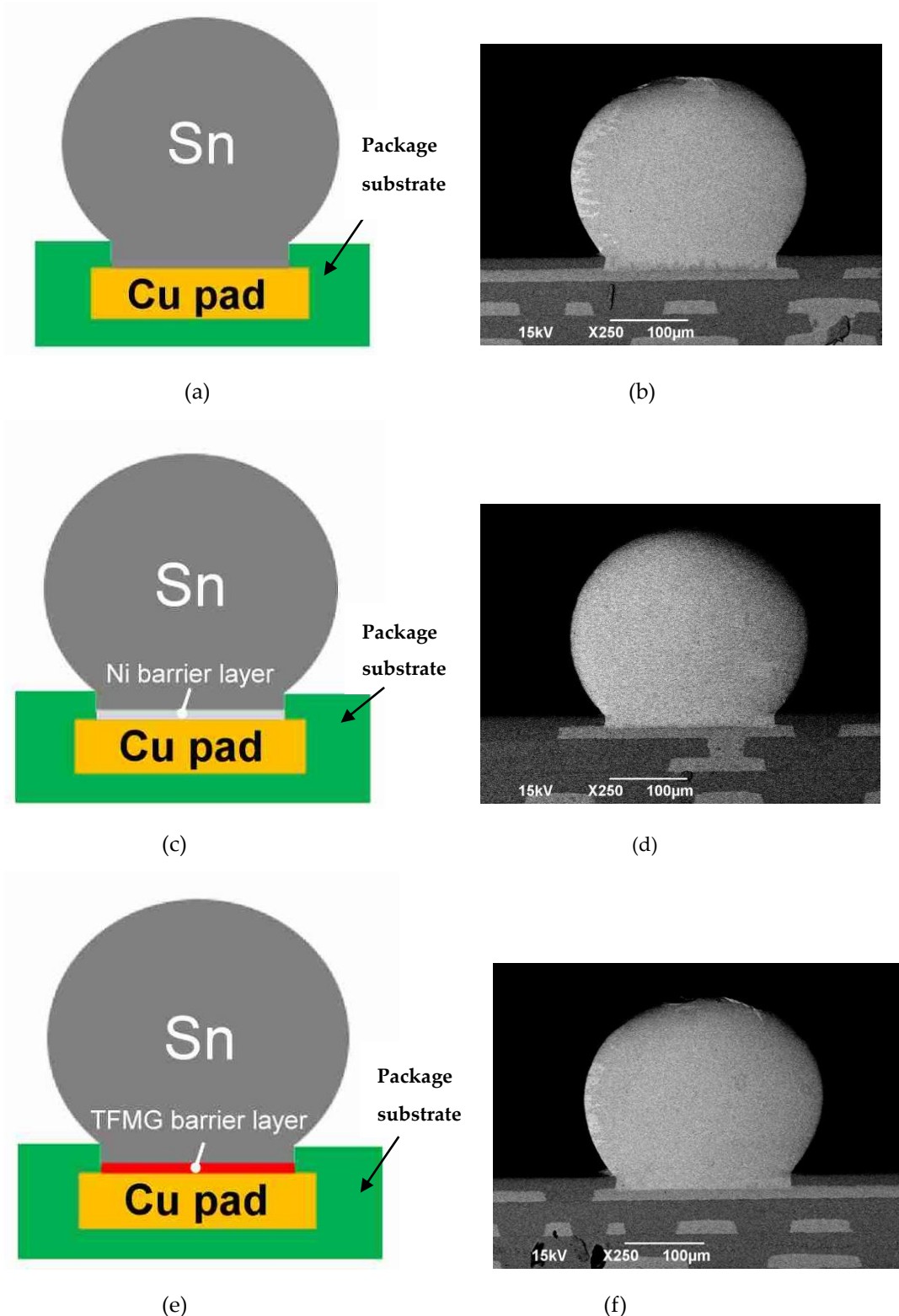

**Figure 7.** Schematic diagram and SEM image of full solder ball without a barrier layer (**a**,**b**), with an Ni barrier layer (2.5 μm thick) (**c**,**d**), and with a $Zr_{53}Cu_{30}Ni_9Al_8$ thin film metallic glass (TFMG) barrier layer (100 nm thick) (**e**, **f**).

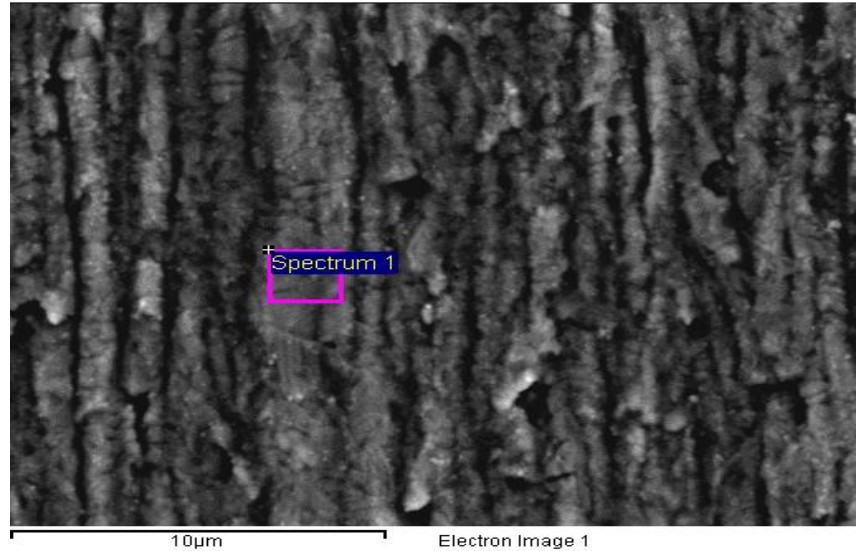

(a)

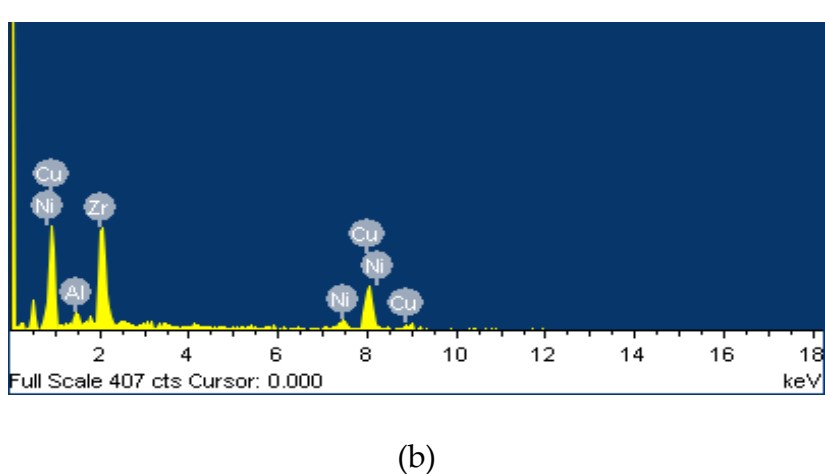

(b)

**Figure 8.** Elemental composition data on the amorphous $Zr_{53}Cu_{30}Ni_9Al_8$ TFMG, identified by SEM from electron image (**a**) and the data of EDS (**b**).

**Table 2.** Elemental composition data on amorphous $Zr_{53}Cu_{30}Ni_9Al_8$ TFMG.

| Element | Weight% | Atomic% |
|---------|---------|---------|
| Al | 2.40 | 6.62 |
| Ni | 8.14 | 10.04 |
| Cu | 31.44 | 32.50 |
| Zr | 58.02 | 50.84 |
| Totals | 100.00 | |

## 3. Results and Discussion

### 3.1. Simulation Results

3.1.1. Radial Distribution Function (RDF)

To analyze the structural differences for different compositions and quenching rates, the RDF curves for all configurations were compared. The results indicated that changes in quenching rates have only slight effects on the structure rearrangement when $Zr_{53}Cu_{30}Ni_9Al_8$ alloy is quenched from 2000 K down to 200 K. This is due to the $Zr_{53}Cu_{30}Ni_9Al_8$ alloy containing four elements, with differences in the atomic radii, which satisfies the Inoue group argument [42]. In addition, it can be seen from Figure 9 that, as the quenching rate went from 0.25 to 25 K/ps, the amorphous ratio of the samples was high. In other words, the $Zr_{53}Cu_{30}Ni_9Al_8$ alloy has a high probability of forming as an amorphous state in this quenching rate interval. Figure 10 shows that, as the temperature was decreased from 1800 K to 200 K, the first peak of the RDF curve increased while the full width at half maximum (FWHM) decreased gradually. For instance, as shown in Figure 10a, the height of the first peak under a quenching rate of 0.25 K/ps increased from 2.2 to 2.8, while the FWHM decreased from 1.9 nm to 1.4 nm as the temperature decreased from 1800 K to 200 K. Same situations can be found for the quenching rates of 2.5 K/ps and 25 K/ps, as shown in Figure 10b,c, respectively. This indicates that the density of atoms was distributed at higher concentration as a function of distance from a reference particle at lower temperatures, exhibiting a more crystalline state, especially when comparing the glassy and the liquid state. In addition, the second peaks of RDF curves became more explicit, splitting as the temperature decreased, which is a common feature of amorphous metal. Therefore, the results confirmed the structural change of microstructures as being in short-range order under the fast quenching rates. For the total pair, the first splitting occurred at about 500 K with a higher quenching rate, but the first splitting occurred at about 300 K with a lower quenching rate, below the temperature of $T_g$. $T_{split}$ is determined by visual inspection of the RDF curve at different temperatures. The results revealed that some substructures formed in like atom pairs before reaching the final glassy state.

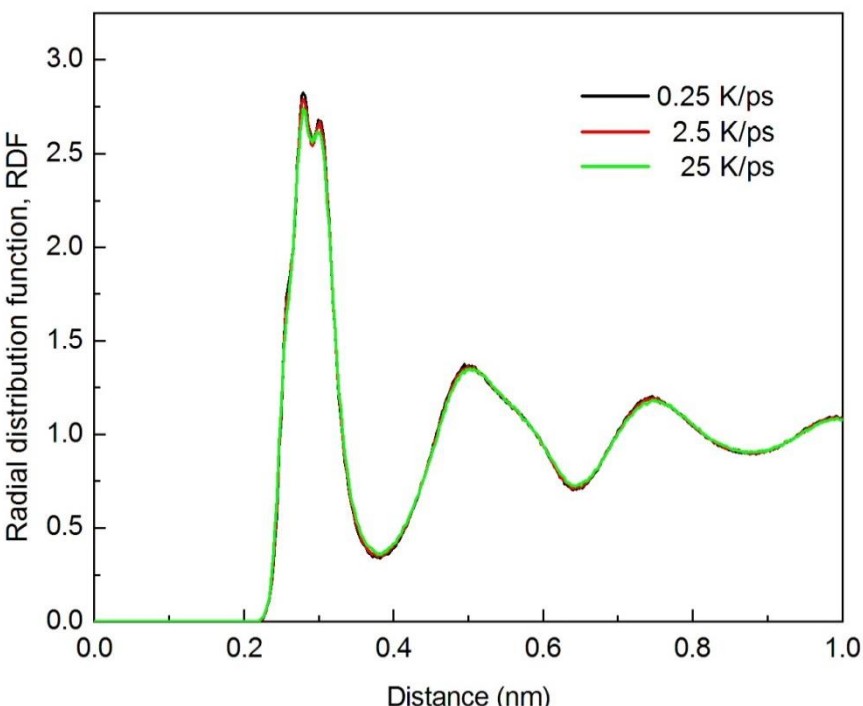

**Figure 9.** Radial distribution functions (RDFs) of $Zr_{53}Cu_{30}Ni_9Al_8$ at 100 K with various quenching rates.

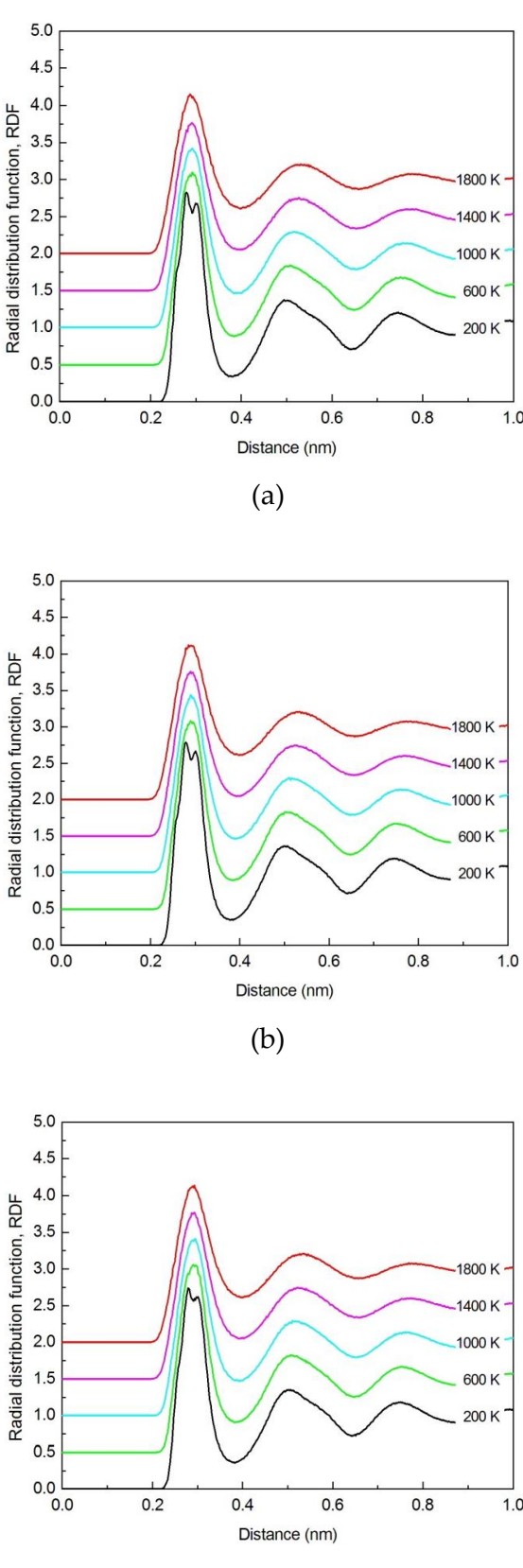

**Figure 10.** RDFs of $Zr_{53}Cu_{30}Ni_9Al_8$ at various temperatures with quenching rates of (**a**) 0.25 K/ps, (**b**) 2.5 K/ps, and (**c**) 25 K/ps.

### 3.1.2. Honeycutt–Anderson (HA) Bond Pair Analysis

To obtain a three-dimensional description of the local atomic configuration of $Zr_{53}Cu_{30}Ni_9Al_8$, the Honeycutt–Anderson (HA) approach [41] is considered to be capable of analyzing in detail the transformation between local structures at different temperatures. According to the definition of the HA bond-type index, pairs of atoms under consideration in a system can be described by four numbers *ijkl*. The first integer *i* of 1 denotes that they are bonded in the root pair; otherwise, *i* is 2. The second integer *j* represents the number of near neighbors shared in common by the two given atoms. The third integer *k* is the number of bonds among the shared neighbors. These three numbers are not sufficient to characterize a diagram uniquely; thus, a fourth integer *l* is added to resolve the ambiguity with respect to the arrangement of the atomic bonds. Applying this HA bond-type index, the bond types between two atoms can be determined clearly. The HA indexes of 1421 and 1422 represent face-centered cubic (FCC) and HCP crystal structures, respectively, and those of 1431, 1541, and 1551, which occupy the largest fraction in the amorphous or liquid state, are used to search the icosahedral local structures. The 1551 pair is particularly characteristic of the icosahedral ordering; 1541 and 1431 are indexes for the defect icosahedra and FCC defect local (or distorted icosahedra) structures, respectively. HA indexes 1661 and 1441 are employed to identify the local body-centered cubic (BCC) structure. Finally, the indexes 1321 and 1311 represent the packing related to rhombohedral pairs that tend to evolve when the 1551 packing forms, which can be viewed as the side product accompanying icosahedral atomic packing. Figure 11 shows the HA index distribution of $Zr_{53}Cu_{30}Ni_9Al_8$ under different quenching rates, where the fraction of icosahedra-like local structures (1551, 1541, and 1431) is over 70%. The fractions of these three icosahedra-like structures are very close, and each of them occupies about 25% of all HA fractions. For other HA indexes, the HCP local structure (1422), BCC local structures (1441 and 1661), and FCC local structure (1421) constitute about 12.5%, 2.5%, 5.2%, and 5.9%, respectively. The HA index distribution of icosahedra-like structures verifies the amorphous structure, which is consistent with the HA analysis results reported previously for MGs [43].

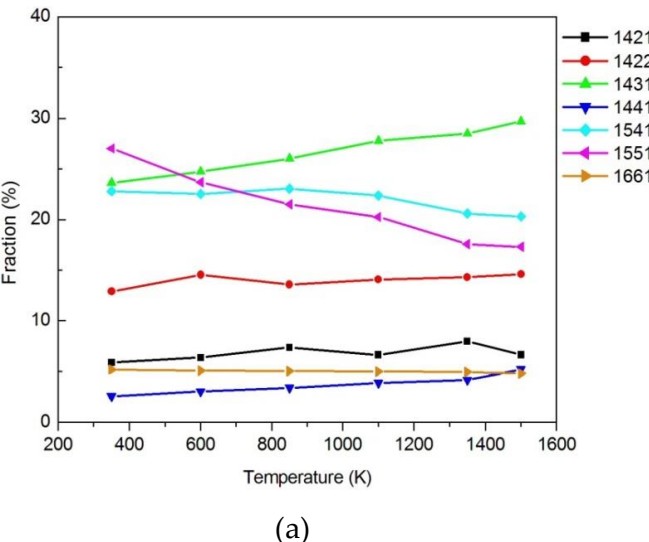

(a)

**Figure 11.** *Cont.*

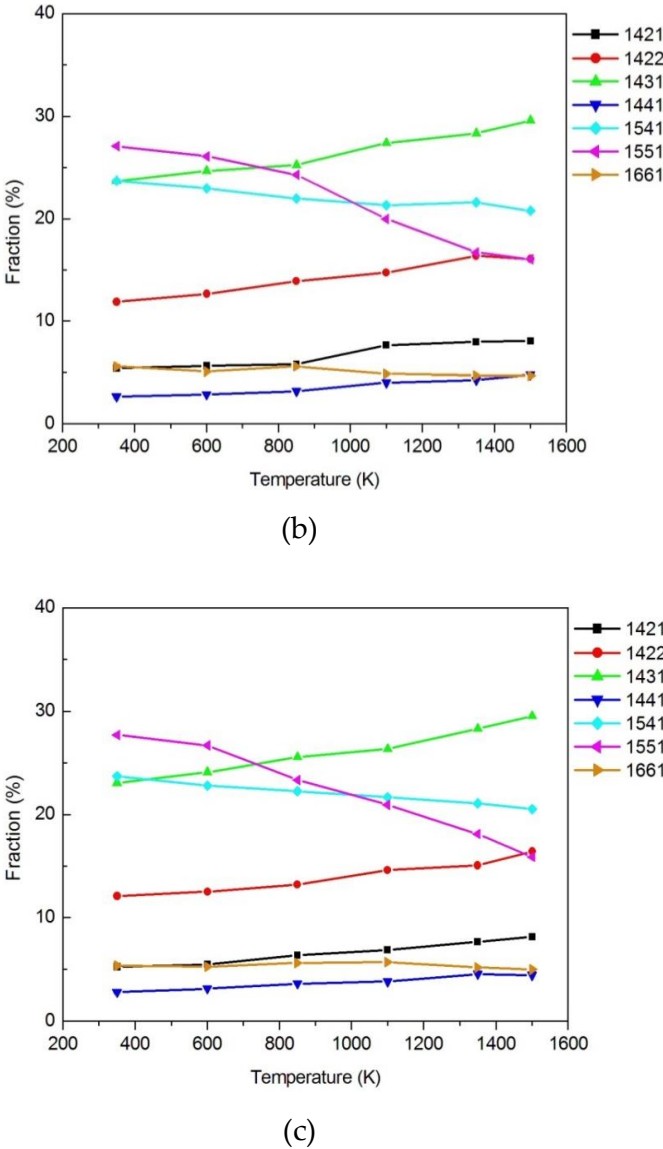

**Figure 11.** The variations of the Honeycutt–Anderson (HA) indexes for the $Zr_{53}Cu_{30}Ni_9Al_8$ alloy quenched at (**a**) 25K/ps, (**b**) 2.5K/ps, and (**c**) 0.25K/ps at different temperatures during quenching process.

The atomic radii of Zr, Cu, Ni, and Al are 1.55, 1.35, 1.35, and 1.25 Å, respectively, with the atomic size of Al smaller than that of Zr by about 20%, while the HA fraction distributions for different atom type pairs maybe different. For this reason, it is necessary to further explore the relationship between different atom pairs and the HA index distribution. Therefore, a more detailed analysis of the HA indexes of different atomic pairs in $Zr_{53}Cu_{30}Ni_9Al_8$ after quenching of 25 K/ps at 100 K is shown in Figure 12. The Zr-related HA indexes (Zr–Zr, Zr–Cu, Zr–Ni, and Zr–Al) form the highest fractions of 1541 and 1431 among all pair types, where 1541 and 1431 are indexes for the defect icosahedra and FCC defect local (or distorted icosahedra) structures, respectively. The Cu, Ni, and Al-related HA indexes (Cu–Cu, Cu–Ni, Cu–Al, Ni–Al, Ni–Ni, and Al–Al) form the highest fractions of 1441 among all pair types, which are employed to identify the local BCC structure. This indicates that the zirconium atoms more easily produce an amorphous state with other atoms due to a larger difference in atomic radius.

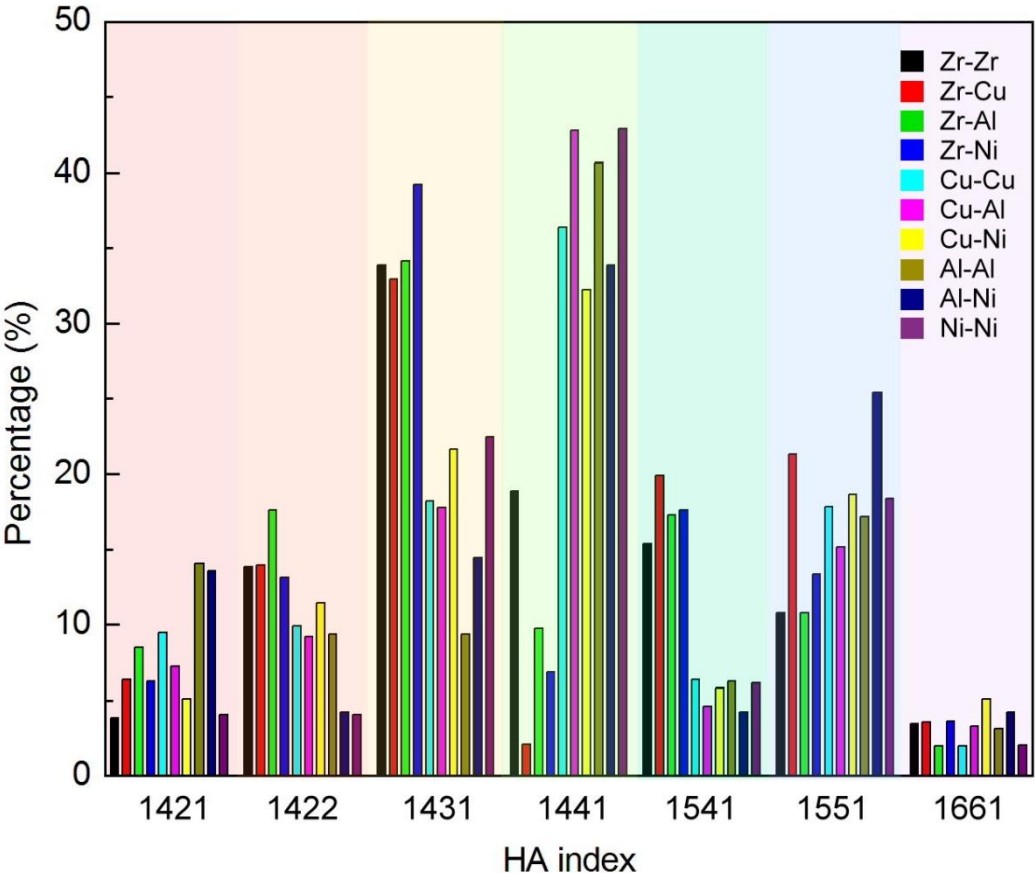

**Figure 12.** HA indexes for different pairs of the $Zr_{53}Cu_{30}Ni_9Al_8$ MG (quenching rate of 25 K/ps and 100 K).

### 3.1.3. Diffusion Between Zr–Cu–Ni–Al and Cu

In order to understand the diffusion behavior between copper and various structures of Zr–Cu–Ni–Al alloys, the interface between Cu and Zr–Cu–Ni–Al alloy layers was investigated at various time steps. Figure 13 shows the cross-section snapshots of the Cu/Zr–Cu–Ni–Al interface under quenching rates of 0.25, 2.5, and 25 K/ps at 600 K after a 2000-ps equilibration process. As demonstrated in this figure, no obvious proliferation was found of Zr–Cu–Ni–Al with quenching rates between 0.25 K/ps and 25 K/ps, while only some of the Cu atoms deviated from their original lattice positions. However, as the time increased, the thickness of the inter-diffusion layer (also called the interfacial region), where the individual concentrations of Cu and Zr–Cu–Ni–Al were both over 5 at%, became thicker. To further quantitatively characterize the diffusion process, the concentration profile was acquired. Figure 14 shows the concentration distributions of Cu and Zr–Cu–Ni–Al along the diffusion couple (Z-axis) direction obtained at 600 K after 2000 ps with different quenching rates. The thickness of the interfacial region was only approximately 1 nm when the quenching rate ranged between 0.25 K/ps and 25 K/ps. This indicates that the effects of the three different quenching rates on the diffusion are very small since the microstructure became highly amorphous (>98%) at these quenching rates. The mean square displacement (MSD) of atoms in the interfacial region was used to investigate their dynamical properties. Figure 15 shows the MSD of Cu and Zr–Cu–Ni–Al as a function of time within the temperature range from 500 K to 900 K obtained in the present work. It is clear that the slopes of the MSD profile are generally larger at a higher temperature. However, the effects of the three different quenching rates on the diffusion are also very small. According to the different temperatures and the MSD curves of the Einstein equation, the total diffusion coefficient near the Cu/Zr–Cu–Ni–Al interface at different temperatures could be obtained as shown in Figure 16. It can be inferred that

the diffusion coefficients between Cu and Zr–Cu–Ni–Al significantly increased with the increase in temperature when the system temperature exceeded the glass transition temperature. The values of these diffusion coefficients had no significant difference at different quenching rates, indicating that there was no significant difference in structure at these quenching rates.

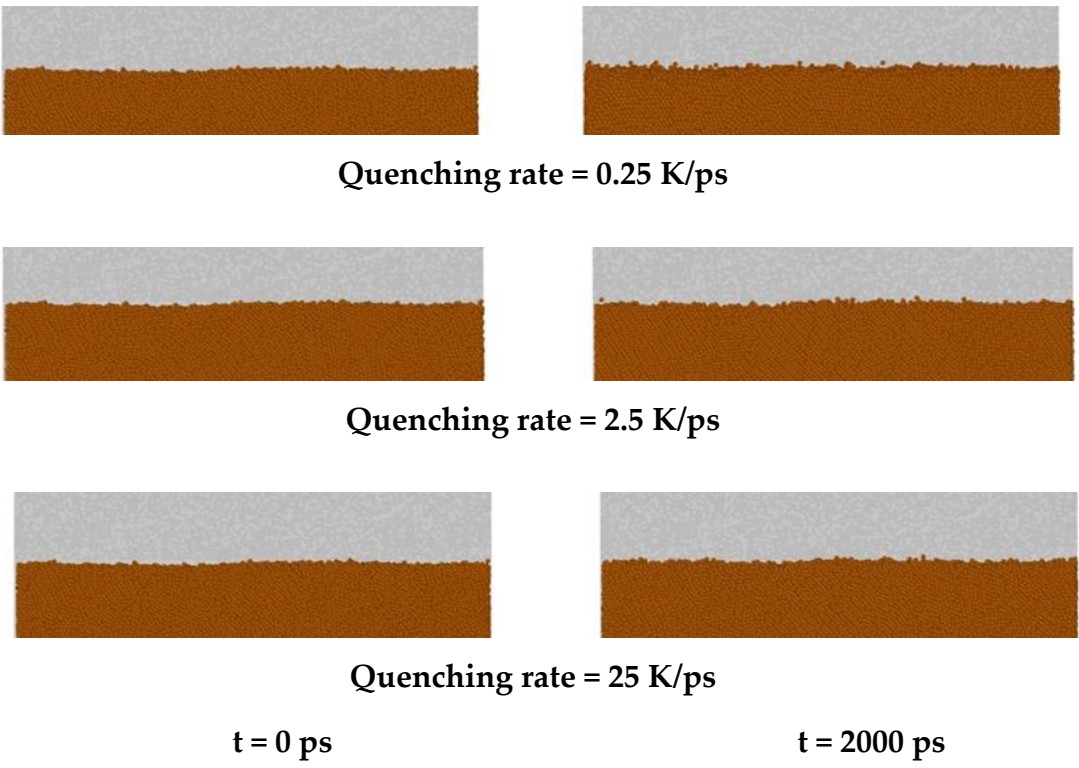

**Figure 13.** Cross-section of $Zr_{53}Cu_{30}Ni_9Al_8$/Cu interface with $Zr_{53}Cu_{30}Ni_9Al_8$ alloy under quenching rates of 0.25, 2.5, and 25 K/ps at 600 K after a 2000-ps equilibration process (Cu atoms in the polycrystalline layer are colored in orange, while the atoms of $Zr_{53}Cu_{30}Ni_9Al_8$ are colored in gray).

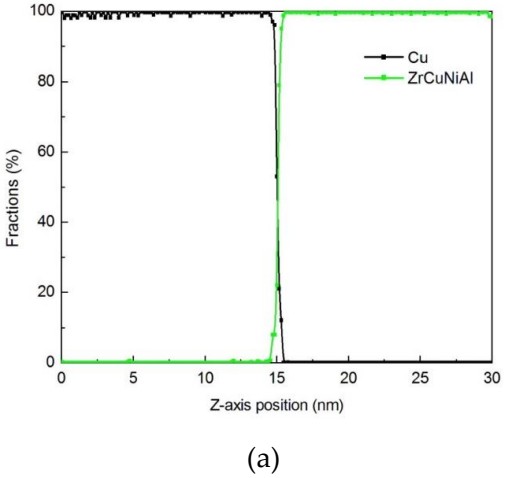

(a)

**Figure 14.** *Cont.*

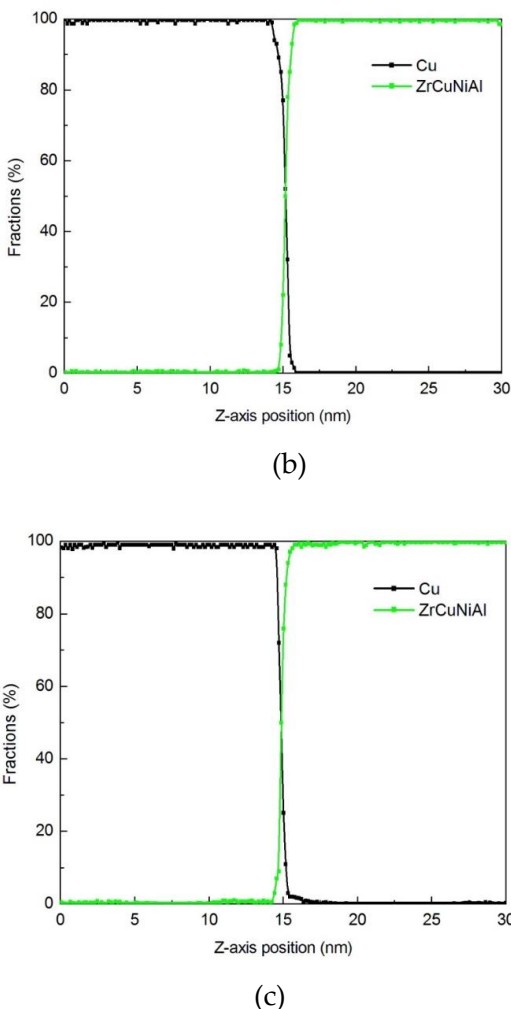

(b)

(c)

**Figure 14.** Concentration distributions of $Zr_{53}Cu_{30}Ni_9Al_8$/Cu quenched at (**a**) 25 K/ps, (**b**) 2.5 K/ps, and (**c**) 0.25 K/ps along the diffusion couple (Z-axis) direction (at 600 K after a 2000-ps equilibration process).

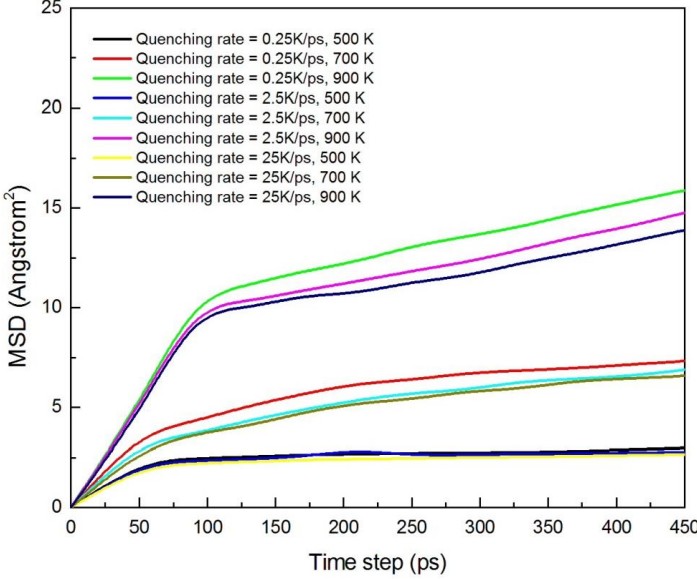

**Figure 15.** Mean square displacement (MSD) of Cu and Zr–Cu–Ni–Al as a function of time at various temperatures and different quenching rates.

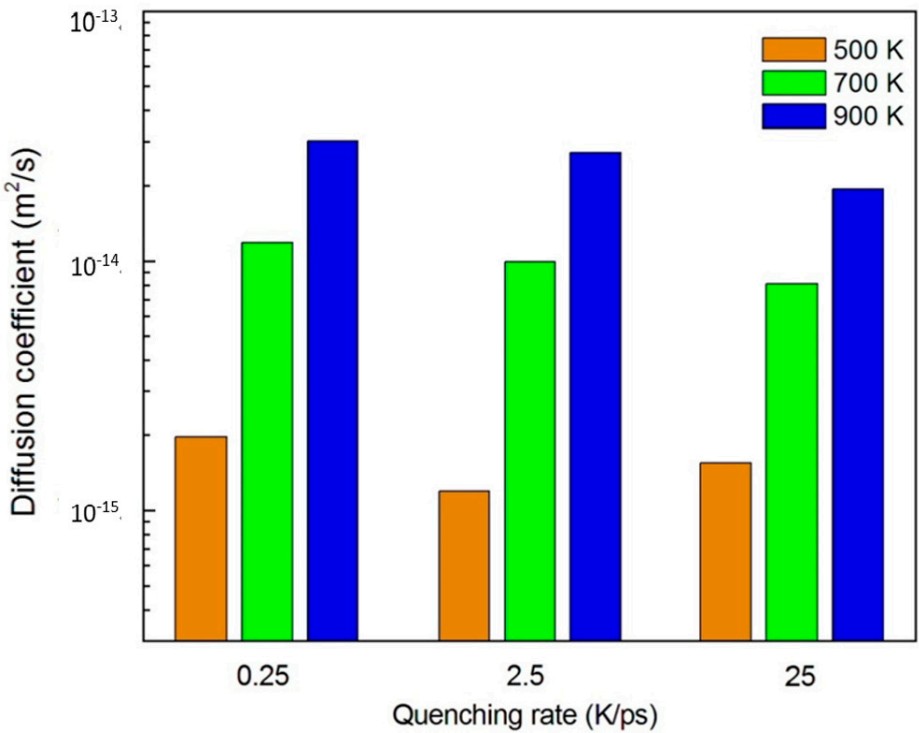

**Figure 16.** Diffusion coefficient between Cu and Zr–Cu–Ni–Al at different quenching rates.

### 3.1.4. Tensile Behavior

In order to explore the mechanical behavior of the amorphous state alloy, Zr–Cu–Ni–Al MGs at three different quenching rates were exerted by tensile deformation. It was found that the effects of the three different quenching rates on the mechanical behavior under tensile test were very small. Henceforth, we only show the tensile results with the quenching rate at 25 K/ps of Zr–Cu–Ni–Al. Figure 17A represents the stress–strain curve of the bilayer material composed of Cu and Zr–Cu–Ni–Al MG under tensile loading of the perfect interface model at a temperature of 100 K and strain rate of $1 \times 10^{10}$ %/s. It can be seen that the tensile stress increased gradually with increasing strain from (a) to (b) and reached a maximum value. After that, the stress suddenly dropped at (c). This was followed by a more steady flow regime during which some serrations are evident from (c) to (e). The sudden drop was caused by the formation of voids arising from the Zr–Cu–Ni–Al/Cu interface. However, for samples produced with a short quenching time, the stress–strain curves were smoother and the behavior was close to the ideal elastic–perfectly plastic response. This phenomenon of sudden drop in stress is the combined effect of dislocations undergoing slippage along the slip plane and void nucleation in the polycrystal copper. The void nucleation is attributed to the fact that there exist more defects in this local region. They are generally formed by the coalescences of the free volume present in the crystal metallic, and they become the site of weakness during tensile deformation. Figure 17B shows (I) the atomic position, (II) common neighbor analysis (CNA), and (III) local atomic strain snapshots of the interface model captured at different strains until fracture. The extension of the model began with an elastic deformation from its initial state to the first yield state. The initial position of the void occurred in polycrystal Cu, which indicates that the Cu/Zr–Cu–Ni–Al interface and Zr–Cu–Ni–Al MG are stronger than polycrystal Cu. In addition, from this figure, the local strain almost occurred concentrically in the polycrystal Cu layer. On the contrary, the distributions of strain in the Zr–Cu–Ni–Al layer were much more uniform and smaller than in the polycrystal Cu layer.

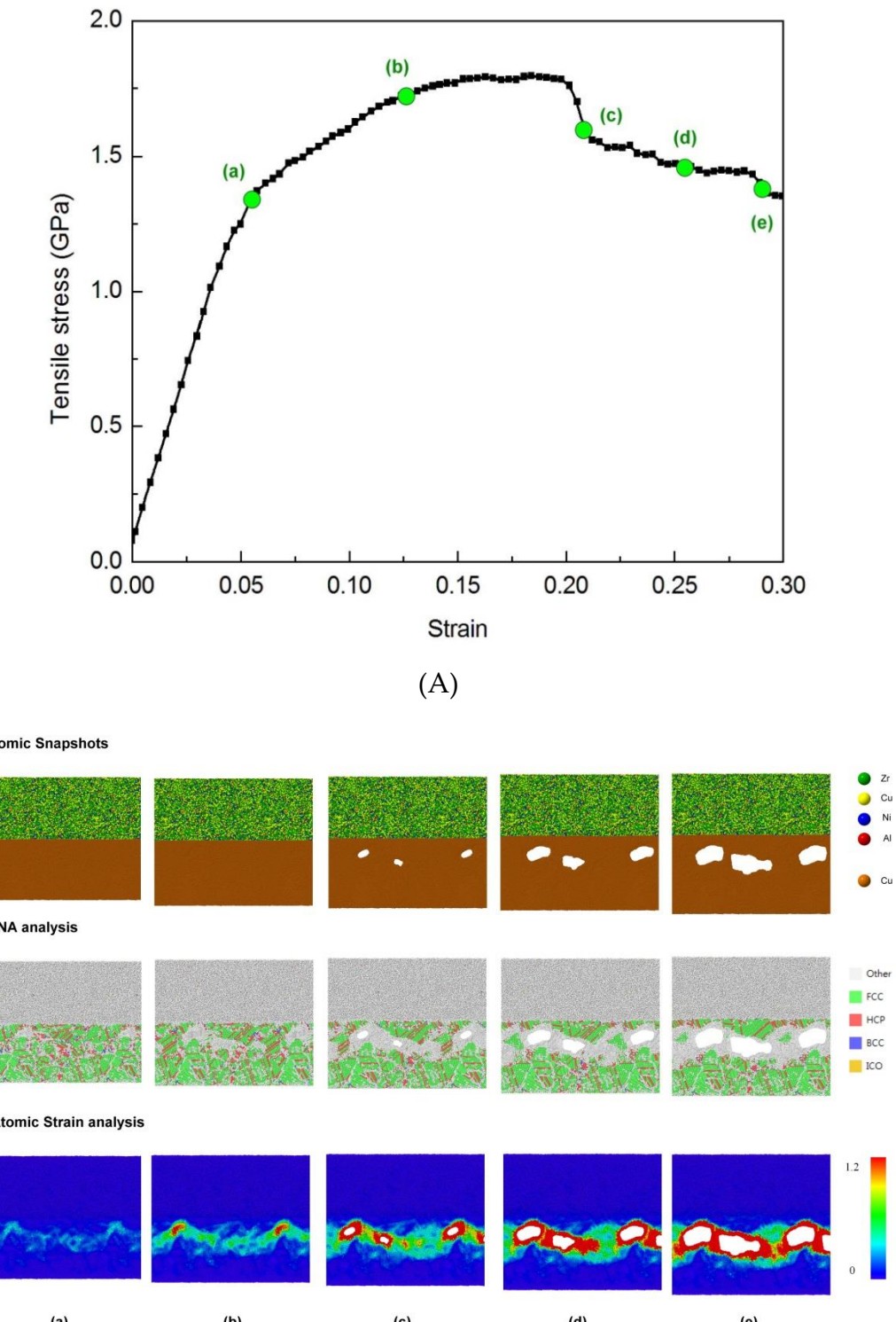

**Figure 17.** (**A**) Stress–strain curve of the Cu/Zr$_{53}$Cu$_{30}$Ni$_9$Al$_8$ interface model under mode-I loading. (**B**) Atomic position snapshots (I), CNA snapshots (II), and atomic strain snapshots (III) of the interface captured under quenching of 25 K/ps at different strains.

### 3.2. IMC Formation with Different Barrier Layers Via Experiment

Figure 18 shows the cross-section morphology of the interface between the Cu and Sn layers in the samples with and without the TFMG barrier layer just after solder joining and after aging at 125 °C for 500 h. In the sample without the barrier layer, bilayer IMCs with a thick scallop-like $Cu_6Sn_5$ of 7–8 μm close to the Sn matrix and a thin $Cu_3Sn$ of 0.5 μm close to the Cu substrate were formed at both interfaces after solder joining. These experimental results are in good agreement with the data in the literature [44]. When aged at 125 °C after 500 h, the interfacial $Cu_6Sn_5$ and $Cu_3Sn$ IMCs grew to 9–10 μm and 0.5–1 μm in thickness, respectively. In the sample with an Ni barrier layer, it can be found that relatively smooth IMCs with thin $Cu_6Sn_5$ of 1–2 μm appeared at interfaces between the Sn and Ni barrier layer after solder joining. When aged at 125 °C after 500 h, the interfacial $Cu_6Sn_5$ IMCs grew to 3–4 μm. In the sample with the TFMG barrier layer, no IMC was observed, whether after solder joining or after aging at 125 °C for 500 h. These data reveal that a thin TFMG barrier layer can still effectively block the subsequent interfacial reactions. Figures 19 and 20 as well as Tables 3 and 4 show a typical EDS spectrum obtained from the IMCs, and the chemical compositions were identified to be $Cu_6Sn_5$ and $Cu_3Sn$, respectively. Furthermore, to protect the surfaces of samples from contamination, it was our standard operation procedure (SOP) that all the surfaces of samples were deposited with a very thin gold layer after the diffusion test. Consequently, some Au materials still remained on the surfaces of the samples and could be detected during the measurement of the SEM/EDS. Consequently, the peak near 2.1 keV was due to Au. These IMCs are the well-known major products that result from Cu/Sn interactions after solid-state aging [45–47]. According to the above observation, the IMC thickness increased with aging time in the samples without a barrier layer and with the Ni barrier layer. In addition, $Cu_6Sn_5$ in the sample without a barrier layer grew much faster than that in the sample with a barrier layer, while the growth of $Cu_3Sn$ was hindered with a barrier layer in this aging condition. It is worth mentioning that, since $Cu_6Sn_5$ and $Cu_3Sn$ are quite brittle, fracture of the long prismatic $Cu_6Sn_5$ and $Cu_3Sn$ IMCs could occur when they suffered from stress concentration and temperature fluctuations.

**Table 3.** Elemental composition data of $Cu_6Sn_5$.

| Element | Weight% | Atomic% |
|---------|---------|---------|
| Cu | 40.51 | 55.99 |
| Sn | 59.49 | 44.01 |
| Totals | 100.00 | |

**Table 4.** Elemental composition data of $Cu_3Sn$.

| Element | Weight% | Atomic% |
|---------|---------|---------|
| Cu | 62.56 | 75.74 |
| Sn | 37.44 | 24.26 |
| Totals | 100.00 | |

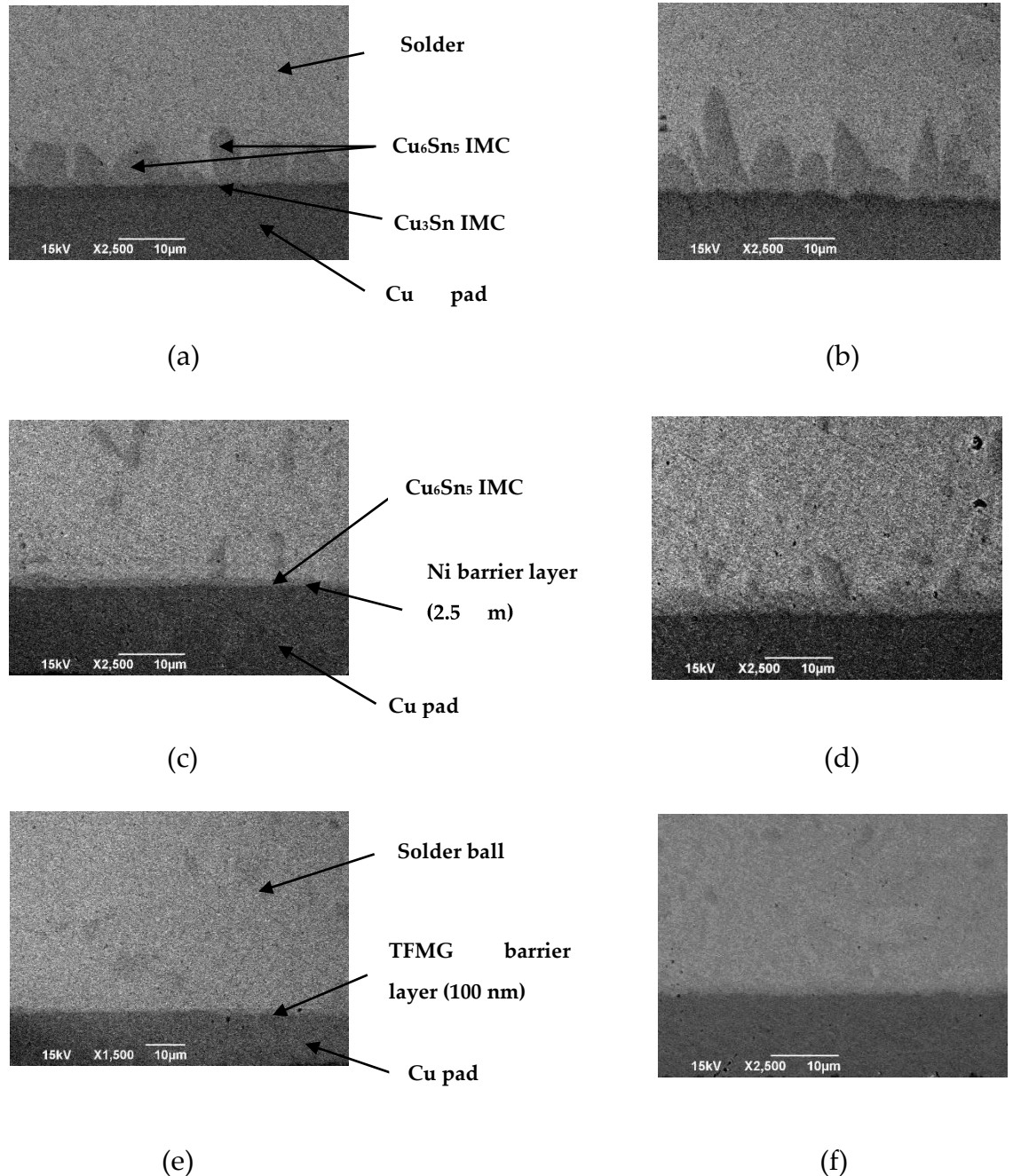

**Figure 18.** SEM image of solder and Cu pad interface without a barrier layer (**a,b**), with an Ni barrier layer (**c,d**), and with a $Zr_{53}Cu_{30}Ni_9Al_8$ TFMG barrier layer (**e,f**) after solder joining (**a,c,e**) and after aging at 125 °C for 500 h (**b**, **d**, **f**).

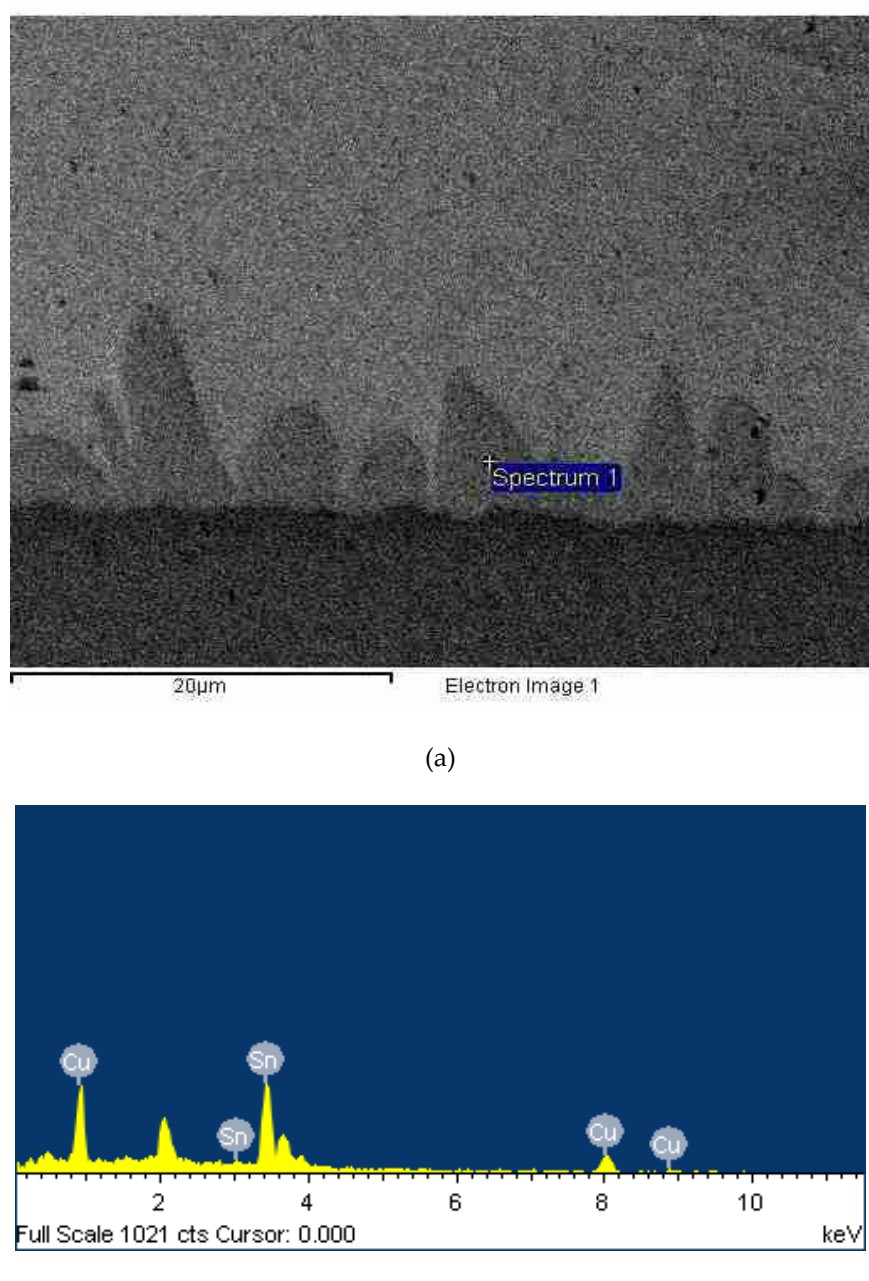

(a)

(b)

**Figure 19.** Elemental composition data of $Cu_6Sn_5$ intermetallic compound (IMC) identified by SEM from electron image (**a**) and the data of EDS (**b**) (from the sample without a barrier layer after aging at 125 °C for 500 h; the peak near 2.1 keV is due to Au).

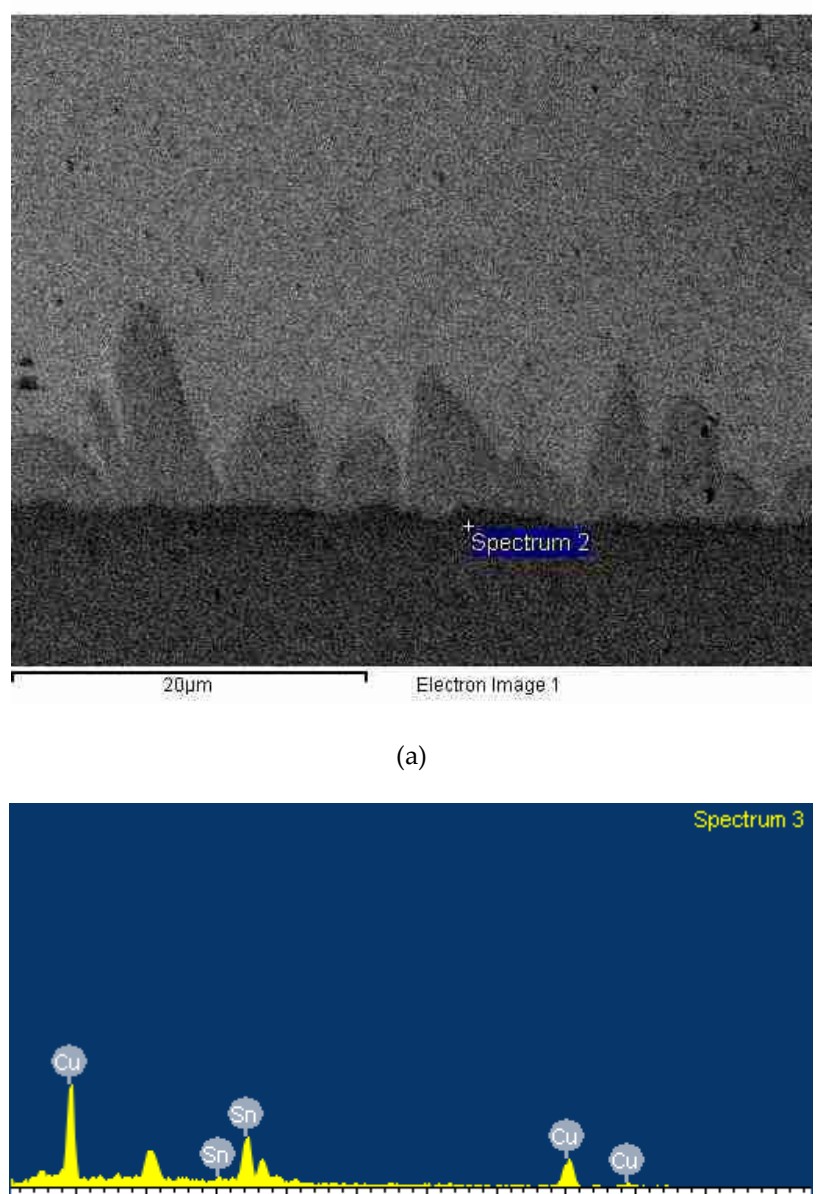

(a)

(b)

**Figure 20.** Elemental composition data of Cu$_3$Sn intermetallic compound (IMC) identified by SEM from electron image (**a**) and the data of EDS (**b**) (from the sample without a barrier layer after aging at 125 °C for 500 h; the peak near 2.1 keV is due to Au).

## 4. Conclusions

The microstructures, diffusion properties, and the strength of the interface between polycrystalline Cu and an amorphous Zr–Cu–Ni–Al thin layer with different quenching rates were simulated using molecular dynamics. Moreover, a diffusion test for three different cases was performed experimentally. The following conclusions can be reached from these simulation and experimental results:

(1) The simulation results of quenching and diffusion tests revealed that the Zr–Cu–Ni–Al amorphous metal exhibits excellent glass-forming ability (GFA) and diffusion barrier performance.

(2) The simulation results of the tensile test showed that all small voids are initiated, propagated, and then coalesced to become large voids in the polycrystal Cu layer only. These large voids

induce highly concentrated strain distributions and finally lead to the rupture of the samples. Meanwhile, the strains induced in Zr–Cu–Ni–Al MG layer are quite uniformly distributed, and no voids are formed on the Cu/Zr–Cu–Ni–Al interface. This implies that the Cu/Zr–Cu–Ni–Al interface and Zr–Cu–Ni–Al MG layer are mechanically stronger than the polycrystalline Cu layer.

(3) The experimental diffusion test revealed that, for the sample of the $Zr_{53}Cu_{30}Ni_9Al_8$ TFMG barrier layer, almost no IMCs were observed, even after aging at 125 °C for 500 h. In other words, the Zr–Cu–Ni–Al amorphous metal exhibits superior diffusion barrier performance compared to the Ni barrier layer, which is in good agreement with the results of the simulation.

**Author Contributions:** P.-H.S. and T.-C.C. conceived and proposed the conceptualization and methodology; P.-H.S. applied the software, performed the simulations and experiments, and drew the figures; P.-H.S. and T.-C.C. performed the validation, formal analysis, data, as well as writing in the stages of original draft preparation, review and editing, and visualization; T.-C.C. conducted the supervision, project administration and funding acquisition. All authors have read and agreed to the published version of the manuscript.

**Funding:** The authors would like to thank the Ministry of Science and Technology of the Republic of China, Taiwan at https://www.most.gov.tw/ for the financial support of this study under Contract No. MOST No. 107-2221-E-006-122- and 108-2221-E-006-191.

**Conflicts of Interest:** The authors declare no conflict of interest.

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
