# Peer review of "Material Properties of Zr–Cu–Ni–Al Thin Films as Diffusion Barrier Layer"

_crystals, doi:10.3390/cryst10060540_

Round 1

Reviewer 1 Report

I thank the authors for addressing my comments adequately. The paper is now much improved and provides a much clearer description of the work and results. 

TMFG acronym is still defined after it's first use in the paper, and there is an issue with the text of "package" in figure 7 which needs to be addressed, but otherwise I have no further comments. 

Author Response

Thanks very much for the comments of the reviewer:

The TMFGs has been defined in its first use in the revised version. Moreover, The description of Fig. 7 has been supplemented in the revised version:

"The schematic deposited structures for three kinds of samples of full solder ball system, called bump, were as shown in Fig. 7. This full solder ball system is generally composed of solder ball, Cu pad, barrier layer, and package substrate colored by green. Fig. 7(a) represents the sample with no barrier layer, while Fig. 7(b) and 7(c) show the sample with Ni- and Zr53Cu30Ni9Al8 TFMG- barrier layers, respectively. "

Reviewer 2 Report

The ms has improved. It is closer to an acceptable form.

There are some  issues that need to be fixed before publication.

1)Equation (1) is not the equation of motion. The authors are just giving an expression for the acceleration (as defined in classical mechanics). In any case, it is really superfluous to write the classical equations of motions (which is text-book knowledge). Please remove eq. 1 and the explanations that follow.

2) On line 112, the authors say that a strain rate of 10x10 s-1 was applied. This is unclear. The strain rate is expressed in percentage/time. Does that mean 10%/s in two directions? Please make clarity.

3) In Fig. 6, the authors should specify in the caption the number of atoms. It does not make sense to say that the alloy have a volume of thousands of nm^3.

Author Response

Thanks very much for the comments of the reviewer.  The replies are as follows:

  1. The Eq. (1) has been removed in the revised version and a short description is supplemented: "In the procedures of numerical simulation, the derivatives in equations of motion were first predicted at time t + Δt by applying Taylor expansions at t. "
  2.  The strain rate has been corrected to " strain rate of 1×1010 %/s" in the revised version.
  3.  The caption of Fig. 6 has been corrected to "Volume variations of 185,000 atoms as a function of temperature under different quenching rates of Zr53Cu30Ni9Al8 alloy" in the revised version.

This manuscript is a resubmission of an earlier submission. The following is a list of the peer review reports and author responses from that submission.

Round 1

Reviewer 1 Report

The authors present a potentially interesting work (also quite well written), but the presentation is poor. For example, I do not see specified anywhere what classical potential and which parameters were used for this multinary system. Another major problem is the presentation of the data. The authors say that they calculated elastic constants by MD and compare them with experimental results. However, it is totally unclear how pure Zirconium was modelled. The authors write only 3 elastic constants (C11, C12, and C44), making one assume that the high-temperature bcc phase was modelled. However, the bcc phase is mechanically unstable at low temperatures (the C11 is larger than C12). If this is fcc Zr, then I really do not undesrtand where the experimental data have been taken from (fcc Zr is not found in nature).

There are too many question marks that make one wonder of the reliability of this work. I recommend rejection. In any case, if the authors feel that they can provide satisfactory clarifications, then I may read more carefully also the rest of the work. Up to now, these are minor and majort problems I found:

1) Define TMFG in the abstract

2) These statements need references: “In the recent years, TiN or Ni, as a barrier layer, has been widely studied and industrially accepted to inhibit rapid copper diffusion in interconnect structures. Unfortunately, TiN and Ni barrier layer are polycrystalline and provide inadequate protection because grain boundaries may presumably serve as fast diffusion paths for copper and could react to form Cu–Sn intermetallics.”

For TiN, I would recommend adding these 3 refs.:

Muehlbacher, et al Cu diffusion in single-crystal and polycrystalline TiN barrier layers: A high-resolution experimental study supported by first-principles calculations, Journal of Applied Physics 118 (2015).

M.N. Popov, et al Point defects at the Σ5 (012)[100] grain boundary in TiN and the early stages of Cu diffusion: An ab initio study, Acta Materialia 144 (2018) 496.

D.G. Sangiovanni, Copper adatom, admolecule transport, and island nucleation on TiN (001) via ab initio molecular dynamics, Applied Surface Science 450 (2018) 180.

3) The authors should add one or two references for Ni as well, to support the statements above.

4) The authors use LAMMPS and Ovito but omit appropriate references:

Plimpton, FAST PARALLEL ALGORITHMS FOR SHORT-RANGE MOLECULAR-DYNAMICS, Journal of Computational Physics 117(1) (1995) 1-19.

Stukowski, Visualization and analysis of atomistic simulation data with OVITO-the Open Visualization Tool, Modelling and Simulation in Materials Science and Engineering 18(1) (2010) 7.

5) How were Cu atoms replaced? Ordered or disordered substitutions? Please specify

6) These two sentences sound redundant, please merge the information:

a series of simulations of heat treatment was conducted within the isothermal isobaric ensemble with an external pressure of zero. The system was initially relaxed under periodic boundary conditions at 300 K for 200 ps within an NPT (constant pressure and constant temperature) ensemble

7) I do not see the time-step used to integrate the equations of motion during MD. Please specify it in the methods section!

8) I do not see anywhere specified what classical potential was used for the simulations, and which parameters have been used to model the intermetallic systems. Add corresponding references in the methods section!!

9) The references are disordered and not clear. For example, Ref 22 at the beginning of page 27. I cannot find the paper by Steinberger in Phys. Rev B (title missing). The authors should take proper care of their reference list.

10) Related to comment 9, how were the elastic constants of Zr in Table 1 calculated? What phase of Zr is this? Note that at ambient conditions, Zr is hcp. Therefore, the elastic constants are more than just C11, C12, and C44. Is this fcc Zr? In that case I really do not understand where the experimental results were taken from, since fcc Zr is most probably never observed in nature. If this is the high-temperature bcc phase of Zr, then the MD simulations should have been performed at the correct temperature to get mechanical stability (meaning C11 > C12). At what temperature were these MD calculations of elastic constants performed??? Note that bcc Zr is mechanically unstable at modestly low temperatures!!!

11) The authors should emphasize that classical models are not always reliable and that they should be subject to careful fitting to first-principles data. See for example these two references and refs. therein.

Edström, et al Large-scale molecular dynamics simulations of TiN/TiN(001) epitaxial film growth Journal of Vacuum Science & Technology A 34, 041509 (2016)

Edström, et al Effects of incident N atom kinetic energy on TiN/TiN(001) film growth dynamics: A molecular dynamics investigation Journal of Applied Physics 121, 025302 (2017)

Reviewer 2 Report

Review of Material Properties of Zr-Cu-Ni-Al Thin Films as Diffusion Barrier Layer

While there is a lot of good and interesting data in this paper, which discusses the use of a ZrCuNiAl diffusion barrier layer to prevent diffusion within interconnect structures, there is a lack of detailed analysis of the data which makes it difficult to recommend for publication in its current form.

Acronyms are not defined throughout, and used in the abstract without definition (e.g. TFMG and IMC in abstract, MD, LAMMPS, OVITO, NPT, RDF, HA etc.)

There appears to be at least two different lists of references used which makes it impossible to determine which is the appropriate one to look at. There appears to be only 31 references in the body of the text, but the reference list goes up to 116. There is also a lack of discussion around recent work on self-forming barrier layers which would be worth noting.

The figure cations are inadequate to describe the figures, as they should be able to be understood without having to refer constantly to the text. There isn’t even any explanation of the techniques used to generate the images.

Figure 7 seems to be identical to figure 17, except (a) and (b) from 17 have been left out of figure 7, to the point that in figure 7 the figures start at (c).

At a number of points in the document silver appears to be used as shorthand for Zr, Ni and Al, without any justification.

On line 67, “IMC is almost found and grown in Sn” … either it is or it isn’t found, but almost found doesn’t make sense.

Line 114, “besides the Zr arom…” There is no clear justification given why Zr behaves differently.

Line 126, “No matter quenching rate…” This is confusing.

Line 148, “magnetron sputter deposition system”…. Please give relevant details of the procedure for deposition, systems used, pressures, times…. Similarly for all other measurement systems, more details are needed. What was type of SEM system, details of acquisitions, how were cross sections prepared… For EDS, how were peaks selected, how was the composition calculated, what was the software used… What was the XRD system used and parameters as this is mentioned for the first time on section 3.2.1, where it is claimed the films are amorphous, but no data is shown. What were the oven conditions during heating at 260°C?

Line 153, the element composition data does not appear to be in figure 7 as stated.

Line 165, What is the Inuoe group argument? Are there meant to be references here? Please elaborate.

Line 171, The FWHM decrease is not obvious to me, how was this determined? Similarly I do not follow how this indicates that the atomic neighbour shells are defined better.

Line 176-177, It states that first splitting occurs at 500 K/300 K but data is only shown for 200 K and 600 K.

Line 178, can the authors elaborate on “somewhat by visual inspection”

Line 182 – 197, there is very little detail on how the HA pair is capable of analysing in detail the transformation. At the very least a few references would be useful. From the figure there appears to be a concomitant decrease in 1551 and increase in 1431 at increasing temperature, is this expected and what does this indicate? There is in general a lack of analysis of the changes presented here other than observations.

Line 213 – 214, “investigated at various time steps.” By what method, and how? What were the time steps? Method not well defined. Not really clear what figure is showing.

Line 218, what do the authors mean by an interdiffusion layer. Is this described by the apparent roughening of the interface in the images and is this really a layer getting thicker or just a change in structure? There are so many figures, would it possible to merge some of them, particularly 12 and 13, which would potentially make things clearer and easier to interpret.

Line 222, is this at 700 K or 600 K, which is used in the caption.  

Line 229, State what temperature is presented.

Line 237, while small it does look like there are obvious trends with different quenching rates, with valuses generally lower with faster quenching rate, apart from 500 K at 2.5. Why is this point different?

Line 242, the effects at the three different rates are not show so this statement is difficult to confirm, particularly since on line 250 it is stated that the stress-strain curves are smoother for short quenching time. This data should be presented, possibly in supplemental info, if available.

Line 248, define serrations.

Line 263, this section contains a lot of repetition from section 2.2. XRD data not shown.

Line 283 – 284, can the regions being described be highlighted in the figures as there are hard to see, particularly the 0.5 um Cu3Sn.

Line 291, “would have little influence” this seems like speculation.

Line 292, What samples are these from? What are the unassigned peaks, particularly the peak at ~2.05 keV? Is this Zr and how would this affect the interpretation in line 283 of the interface composition? Is there any oxygen observed or are cross sections prepared in-situ?

Figure 7, 17, what are green areas?

Figure 8,9, can key features in spectra be labelled or commented on in caption?

Figure 12, would it be more appropriate to just show narrow regions around the interface, rather than the whole cell, so that differences are easier to visualise.

Line 359-360, Can the authors elaborate more on this as I don’t feel this has been adequately explained in the text as is test shown on the Cu interface?

Line 361, in which experiment?